# Chemerin-Induced Down-Regulation of Placenta-Derived Exosomal miR-140-3p and miR-574-3p Promotes Umbilical Vein Endothelial Cells Proliferation, Migration, and Tube Formation in Gestational Diabetes Mellitus

**DOI:** 10.3390/cells11213457

**Published:** 2022-11-01

**Authors:** Lixia Zhang, Qi Wu, Shuqi Zhu, Yibo Tang, Yanmin Chen, Danqing Chen, Zhaoxia Liang

**Affiliations:** 1Department of Obstetrics, Women’s Hospital, School of Medicine, Zhejiang University, Hangzhou 310006, China; 2Department of Epidemiology, School of Public Health and Tropical Medicine, Tulane University, New Orleans, LA 70112, USA

**Keywords:** placenta-derived exosomal miRNAs, chemerin, umbilical vein endothelial cells, gestational diabetes mellitus

## Abstract

Gestational diabetes mellitus (GDM) leads to poor pregnancy outcomes and fetoplacental endothelial dysfunction; however, the underlying mechanisms remain unknown. This study aimed to investigate the effect of placenta-derived exosomal miRNAs on fetoplacental endothelial dysfunction in GDM, as well as to further explore the role of chemerin to this end. Placenta-derived exosomal miR-140-3p and miR-574-3p expression (next-generation sequencing, quantitative real-time PCR), its interactions with cell function (Cell Counting Kit-8, Transwell, tube formation assay), chemerin interactions (Western blotting), and placental inflammation (immunofluorescence staining, enzyme-linked immunosorbent assay) were investigated. Placenta-derived exosomal miR-140-3p and miR-574-3p were downregulated in GDM. Additionally, miR-140-3p and miR-574-3p inhibited the proliferation, migration, and tube formation ability of umbilical vein endothelial cells by targeting vascular endothelial growth factor. Interestingly, miR-140-3p and miR-574-3p expression levels were negatively correlated with chemerin, which induced placental inflammation through the recruitment of macrophage cells and release of IL-18 and IL-1β. These findings indicate that chemerin reduces placenta-derived exosomal miR-140-3p and miR-574-3p levels by inducing placental inflammation, thereby promoting the proliferation, migration, and tube formation of umbilical vein endothelial cells in GDM, providing a novel perspective on the underlying pathogenesis and therapeutic targets for GDM and its offspring complications.

## 1. Introduction

Gestational diabetes mellitus (GDM) is characterized by glucose intolerance at the onset of pregnancy, which is commonly diagnosed in the second or third trimester of pregnancy and is not clearly type 1 or 2 diabetes. GDM affects approximately 15% of all pregnancies worldwide and its incidence is increasing. Notably, GDM confers an increased risk of severe pregnancy complications, including macrosomia, shoulder dystocia, and neonatal hypoglycemia, along with an increased likelihood of long-term offspring metabolic diseases such as type 2 diabetes, obesity, and cardiovascular disease [1,2,3]. Therefore, it is crucial that the underlying molecular mechanisms, prognostic markers, and molecular therapeutic targets be identified to prevent GDM and improve the treatment outcomes for GDM complications.

Recent studies indicate that GDM is associated with fetoplacental endothelial dysfunction [4,5]. The fetoplacental vasculature plays a key role in trans-placental nutrient from the mother to the fetus; changes may lead to abnormal fetal growth and development, and even to alterations in the health status of the newborn, child, and adult. Accordingly, fetoplacental endothelial dysfunction in GDM may contribute to offspring complications; however, its underlying mechanisms remain poorly understood. Studies have shown that oxidative stress and inflammatory activation are related to cardiac and diabetic vascular endothelial dysfunction [6,7]. It has been reported that macrophage infiltration is increased in GDM placenta compared to the normal group [8]. Furthermore, the relative M1 macrophage ratio was higher in GDM placenta, while the M2 macrophage relative ratio is the opposite [9]. Additionally, several studies have documented changes to inflammatory cytokines such as interleukin-8 (IL-8), IL-15, and tumor necrosis factor α (TNF-α) in GDM placentas [10,11]. Placental inflammation has been observed in GDM, and may play a role in fetoplacental endothelial dysfunction [12].

Insulin, adenosine, and adipokine receptors are involved in the fetoplacental vascular dysfunction seen in GDM [13]. Chemerin, a recently discovered adipokine, plays a key role in the inflammatory response and metabolic disease [14]. Additionally, it has been shown that chemerin may affect angiogenesis by stimulating the production of vascular endothelial growth factor (VEGF) [15]. In women with preeclampsia, placental chemerin expression and release were increased, and chemerin overexpression inhibited the migration, invasion, and tube formation of human umbilical vein endothelial cells (HUVECs) [16]. Furthermore, chemerin stimulated M1 macrophage polarization and contributed to preeclampsia [17]. Previous studies have shown that levels of chemerin were higher in the blood, placental tissue, and cord blood of diabetic pregnancy patients [18,19]. Aggregation of chemerin led to macrophage recruitment and the release of inflammatory cytokines in the brains of offspring from chemerin-treated GDM dams [20]. However, whether chemerin has a role in regulating the function of fetoplacental endothelial cells in GDM has yet to be confirmed.

Extracellular vesicles (EVs) are nanosized particles containing bioactive content (including proteins, lipids and nucleic acids), and are enclosed by a lipid bilayer. The syncytiotrophoblast layer of the placenta secretes small EVs (exosomes and microvesicles) into the maternal circulation from as early as six weeks of gestation during pregnancy [21]. It has been shown that placental EVs are bioactive and contribute to the maternal proinflammatory state observed in GDM by stimulating HUVECs to release proinflammatory cytokines [22]. Moreover, human umbilical endothelium-derived exosomes have been reported to contribute to fetoplacental endothelial dysfunction in GDM pregnancies [23]. Exosomes are a subtype of extracellular vesicles that are specifically packaged with signaling molecules, including protein, messenger RNA (mRNA), and noncoding RNA. Exosome secretion has been reported in placental cells as a response to changes occurring across gestation. Placenta-derived exosomes regulate endothelial cell function and potentially affect the development of the maternal–fetal vasculature [24]. The concentration of exosomes from maternal plasma is significantly elevated in GDM, which promotes the release of proinflammatory cytokines from endothelial cells [22]. Extracellular vesicle-associated microRNAs (miRNAs) are altered in GDM in the context of insulin sensitivity [25]. Studies have shown that miRNAs mediate vascular endothelial dysfunction in chronic kidney disease, preeclampsia, and high-glucose conditions [26,27,28]. However, the impact of placenta-derived exosomes and miRNAs on fetoplacental endothelial dysfunction in GDM is not well understood.

In this study, we analyze the effects of placenta-derived exosomal miRNAs on fetoplacental endothelial dysfunction in GDM and the role of chemerin. Placenta-derived exosomal miR-140-3p and miR-574-3p were reduced in chemerin-induced GDM, resulting in abnormal proliferation, migration, and tube formation of umbilical vein endothelial cells.

## 2. Materials and Methods

### 2.1. Patient Recruitment

Pregnant women with GDM and pregnant women with normal glucose levels who delivered by cesarean section were recruited to this study from the delivery suites of the Women’s Hospital School of Medicine at Zhejiang University. GDM was diagnosed as abnormal blood glucose at any point according to the International Association of Diabetes and Pregnancy Study Group (IADPSG) criteria by 75g OGTT, including fasting plasma glucose (FPG) ≥5.1 mmol/L, 1h postprandial plasma glucose (PG) ≥10.0 mmol/L, and 2h-PG ≥8.5 mmol/L. Pregnant women who had experienced any of the following were excluded from the study: multiple pregnancies, assisted reproduction, an infection of the reproductive system during pregnancy, diabetes mellitus before pregnancy, or other pregnancy complications. The clinical characteristics are detailed in Appendix A. This study was approved by the Ethics Committee of the Women’s Hospital School of Medicine, Zhejiang University. All patients signed written informed consent forms.

### 2.2. Mouse Model Establishment

Male and female DBA/2J mice (aged 4–6 weeks) were purchased from Shanghai Jiesijie Laboratory Animals Co., Ltd. (Shanghai, China). All mice were acclimated for one week before treatment. Our experiments were approved by the Institutional Animal Research Committee and Ethics Committee of Zhejiang University. A typical diabetic model was established by intraperitoneal injection of streptozotocin (STZ, 40 mg/kg) for five consecutive days, beginning three days after pregnancy [29]. The chemerin-induced maternal diabetic model was established by intraperitoneal injection of chemerin (3–4 μg/g body weight, 2325-CM, RD, MN, USA) at gestational day (GD) 2, for three consecutive days [20], and the control group by intraperitoneal injection with normal saline at the same time. Both groups were verified by measuring fasting blood glucose (FBG) and fasting blood insulin levels (FBINS) at GD 10.5 and 18.5 using enzyme-linked immunosorbent assay (ELISA) kits (FBG, S0201S, Beyotime, Shanghai, China; FBINS, E-EL-M1382c, Elabscience, Wuhan, China) and the homeostasis model of assessing insulin resistance (HOMA-IR) (Appendix A). HOMA-IR was calculated by (FBINS × FBG)/22.5.

Chemerin-induced maternal diabetic mice were divided randomly into two groups and compared to control mice. Chemerin treatment (chemR23 knockdown) was administered via an intravenous tail injection of 1 × 10^9^ plaque-forming units (pfu) of chemR23-short hairpin RNA (shRNA) lentivirus on GD 10.5. The controls and chemerin-induced diabetic mice were injected with a 1 × 10^9^ pfu lentivirus vehicle. The mice were delivered by cesarean section on GD 18.5 and their placentas were collected for Western blotting and exosome isolation.

### 2.3. Preparation of Placenta-Derived Exosomes from Placenta Tissue and Exosomes from Trophoblast Cells

Placenta tissue was collected during delivery and stored in phosphate-buffered saline (PBS) at 4 °C. The large blood vessels and connective tissue of placental tissue were removed in an ultra-clean workbench. Then, placental tissue was rinsed repeatedly with PBS for several times. Villous tissue was cut into placental explants of 1–2 mm^3^ size and rinsed repeatedly with 1640 medium (PM150110, Procell, Wuhan, China) with 10% fetal bovine serum (FBS, 10099141C, Gibco, Australia) for 3–5 times. Then, they were placed in a 12-well plate, to which was added 1640 medium containing 10% FBS. Placental explants were cultured in an incubator at 37 °C (5% CO2) for 48 h, and the culture medium was collected after 48 h. Exosome-enriched EVs were isolated by sequential centrifugation at 2000× *g* for 30 min at 4 °C and 10,000× *g* for 60 min. The pellets were then resuspended in PBS and washed twice, followed by filtration using a 0.22-mm filter. Exosome markers, including CD63, CD9, and CD81 [30], and a placenta-derived exosomal-specific marker (placental alkaline phosphatase, PLAP) were analyzed by Western blotting (Appendix A). Placenta-derived exosomes were characterized according to their morphology (typically a round or oval shape) and size (80–100 nm) using transmission electron microscopy (Titan, FEI) and nanoparticle tracking analysis (NTA, NanoSight, NS300, Malvern Panalytical, Malvern, UK) (Appendix A). After treating trophoblast cells with interleukin 18 (IL-18, 10 ng/mL, RAB0810, sigma, MO, USA), interleukin 1 beta (IL-1β, 10 ng/mL, HY-P7073, MCE, NJ, USA), or the supernatant from macrophage cells for 24 h, we collected the culture medium and isolated the exosomes. 

### 2.4. Cell Culture and Treatment

Human umbilical vein endothelial cells (HUVECs) and mouse umbilical vein endothelial cells (MUVECs) were cultured in Dulbecco’s modified Eagle’s medium (DMEM, PM150210, Procell) supplemented with 10% FBS and 100 U penicillin-streptomycin (PB180120, Procell) at 37 °C with 5% CO2. HUVECs and MUVECs were incubated with placenta-derived exosomes or transfected with miRNA agomir or antagomir. Placental macrophages were isolated by collagenase (17100017, Gibco)-trypsin (15050057, Gibco)-DNase (EN0521, Thermo Fisher Scientific, MA, USA) sequential digestion, mesh filtration, red blood cell lysis (11814389001, Roche, Rotkreuz, Switzerland), and centrifugation with lymphocyte separation medium (abs930, Absin, Shanghai, China), and were then cultured in 1640 medium supplemented with 20% FBS at 37 °C with 5% CO2 for 48h until the degree of cell fusion was greater than 80%.

Trophoblast cells were isolated by collagenase-trypsin-DNase sequential digestion, mesh filtration, and centrifugation with Percoll separation medium (P4937, Sigma) and then cultured in DMEM supplemented with 10% FBS, 100 U penicillin-streptomycin, 2 mM L-glutamine (59202C, Sigma), 10 mM sodium pyruvate (P2256, Sigma), and 1% non-essential amino acids (M7145, Sigma) at 37 °C with 5% CO2. Trophoblast cells were co-cultured with supernatant from macrophages and treated with IL-18 and/or IL-1β. Cells were harvested and assays were conducted. 

### 2.5. Small RNA Sequencing

Next gene sequencing was performed according to the manufacturers’ instructions from Illumina. Small RNA sequencing libraries were prepared using TruSeq Small RNA Sample Preparation Kits (RS-200-0012, Illumina, San Diego, CA, USA). After library preparation, the constructed libraries were sequenced using an Illumina Hiseq2000 with single-ended 1X50bp read length.

### 2.6. Real-Time Reverse Transcription Polymerase Chain Reaction (qRT-PCR)

Total RNA was extracted using a miRNeasy Kit. miRNA expression was quantified using specific primers designed for miR-140-3p, pre-miR-140-3p, pri-miR-140-3p, miR-574-3p, pre-miR-574-3p, and pri-miR-574-3p, with a universal reverse sequence primer. For RNA analysis, cDNA was synthesized using a high-capacity cDNA reverse transcription kit (4368814, Applied Biosystems, MA, USA). Quantitative real-time polymerase chain reaction (qPCR) was performed using SYBR green qPCR master mix with specific primers. Relative gene expression was normalized to the expression of the housekeeping gene (U6). The primer sequence was as follows: miR-140-3p loop 5′-GTCGTATCCAGTGCAGGGTCCGAGGTATTCGCACTGGATACGACTGTGGGTG-3′ and forward 5′-TGCGCTACCACAGGGTAGAAC-3′; pri-miR-140-3p forward 5′-CTCTGCATCGAAGGACTCCA-3′and reverse 5′-CCCACCCAATAGACGCCTTA-3′; pre-miR-140-3p forward 5′-CCTGCCAGTGGTTTTACCCT-3′ and reverse 5′-CCTGTCCGTGGTTCTACCCT-3′; miR-574-3p loop 5′-GTCGTATCCAGTGCAGGGTCCGAGGTATTCGCACTGGATA-CGACCCGTGGTT-3′ and forward 5′-TGCGCTACCACAGGGTAGAAC-3′; pri-miR-574-3p forward 5′-GTGTGTGTGAGTGTGTGTCG-3′ and reverse 5′-TGGGACACTTGGGGAAACTT-3′; pre-miR-574-3p forward 5′-TGCGGGCGTGTGAGTGTGTG-3′ and reverse 5′-CGTGCGGGCGTGTGGGTGTG-3′; U6 forward 5′-CGCTTCGGCAGCACATATAC-3′ and reverse 5′-AAATATGGAACGCTTCACGA-3′.

### 2.7. Immunofluorescence Staining

Placental tissue was embedded in OCT, rapidly frozen in liquid nitrogen, and stored at −80 °C. The embedded tissue was cut into 8–10-μm sections, which were fixed and rinsed in acetone and PBS, respectively. After blocking with goat serum, the sections were incubated with primary antibodies against chemerin (Ab72965, 1:200; Abcam, Cambridge, UK) and F4/80 (DF2789, 1:200; Affinity Biosciences, Cincinnati, OH, USA) followed by species-specific secondary antibodies ((Goat Anti-Rabbit IgG Alexa Fluor594-conjugated antibody, S0006, 1:200; Affinity Biosciences) or (Goat Anti-Mouse IgG FITC-conjugated antibody, S0007, 1:200; Affinity Biosciences)). HUVECs were sorted and seeded on coverslips, fixed in 4% paraformaldehyde, and permeabilized with 0.5% Triton X-100. The coverslips containing the cells were incubated with PKH26-Exo. Nuclei were counterstained with diaminobenzene. The cells/sections were examined under a fluorescence microscope.

### 2.8. Western Blotting

HUVECs, MUVECs, or trophoblast cells were washed three times with cold PBS and lysed in radioimmunoprecipitation assay lysis buffer supplemented with a protease inhibitor cocktail. After lysing for 30 min, cell lysates were centrifuged for 30 min at 4 °C and 12,000× g. The protein concentration was quantified using the BCA protein assay kit (P0010S, Beyotime) with 40 μg of protein per sample loaded onto gels. The proteins were separated by sodium dodecyl sulfate–polyacrylamide gel electrophoresis (SDS-PAGE) and transferred onto a polyvinylidene fluoride (PVDF) membrane (IEVH85R, Millipore, MA, USA). The membranes were incubated with primary antibodies for 2 h at room temperature or overnight, followed by exposure to horseradish peroxidase-conjugated anti-IgG secondary antibodies (BA1051, BA1054; Boster Biological Technology Co. Ltd., CA, USA) for 1.5 h. The gray values of the targeted protein bands were detected using ImageJ software (NIH, Bethesda, MD, USA) and the ratio with glyceraldehyde 3-phosphate dehydrogenase was calculated. The primary antibodies used were the following: chemR23 (ab230442, Abcam), NLRP3 (15101, CST, BSN, USA), Asc (DF6304, Affinity), caspase3 (ab32351, Abcam), caspase9 (ab202068, Abcam), pro-caspase1 (83383, CST), pro-IL-1β (12242, CST), pro-IL-18 (M156-3, MBL, Tokyo, Japan), cleaved caspase1 (AF4005, Affinity), IL-1β (AF4006, Affinity), IL-18 (DF6252, Affinity), VEGF (ab52917, Abcam), hnRNP L (ab264340, Abcam), Drosha (DF8601, Affinity), DGCR8 (DF2917, Affinity), Exportin-5 (DF13416, Affinity), CD63 (AF5117, Affinity), CD9 (DF6565, Affinity), CD81 (DF2306, Affinity), PLAP (ab133602, Abcam), and GAPDH (ab8245, Abcam).

### 2.9. Enzyme-Linked Immunosorbent Assay

For assessment of placental inflammation, placental tissue was removed from the pregnant women and mice through dissection and homogenized. The homogenates were analyzed using ELISA kits (IL-18, ml002294, mlbio, Shanghai, China; IL-1β, PI301, Beyotime). All ELISA measurements were performed according to the manufacturer’s instructions.

### 2.10. Cell Counting Kit-8 Assay

After attachment, HUVECs or MUVECs were treated with 100nM miR-140-3p and miR-574-3p agomir and/or antagomir (hsa-miR-140-3p agomir, miR40004597-4-5, RiboBio, Guangzhou, China; mmu-miR-140-3p agomir, miR40000152-4-5, RiboBio; hsa-miR-574-3p agomir, miR40003239-4-5, RiboBio; mmu-miR-574-3p agomir, miR40004894-4-5, RiboBio; hsa-miR-140-3p antagomir, miR30004597-4-5, RiboBio; mmu-miR-140-3p antagomir, miR30000152-4-5, RiboBio; hsa-miR-574-3p antagomir, miR30003239-4-5, RiboBio; mmu-miR-574-3p antagomir, miR30004894-4-5, RiboBio; agomir NC, miR4N0000001-4-5, RiboBio; antagomir NC, miR3N0000001-4-5, RiboBio), and then transfected with 2 μg hnRNP L plasmid vector or overexpression (OE) plasmid (VB900137-4326dwv, VectorBuilder, Chicago, IL, USA) by Lipofectamine 2000 (L3287, Sigma) and incubated at 37 °C for 24 h. Finally, Cell Counting Kit-8 (CCK-8) working solution was added, followed by incubation for 4 h. The absorbance was measured at 450 nm using a spectrophotometric microplate reader.

### 2.11. Transwell Assay

A 200-μL aliquot of the elicited HUVECs or MUVECs (1 × 10^6^/mL) transfected with the hnRNP L vector or OE plasmid was transferred to the upper chambers in conjunction with placenta-derived exosome transfection with miR-140-3p and miR-574-3p agomir and/or antagomir into the lower chambers. The transwell chambers were taken out and washed with calcium-free PBS 24 h later, then fixed with 70% ethanol. After removing the non-migrating cells from the upper layer, the migrated cells were stained with 0.1% crystal violet and examined under a microscope. The number of cells stained with crystal violet was counted in five randomly selected fields at 400×.

### 2.12. Tube Formation Assay

HUVECs or MUVECs were seeded on growth factor-reduced Matrigel-coated 96-well plates at 3 × 10^4^ cells per well for 24 h. Endothelial tubule formation was observed and photographed using an inverted microscope. Total tube lengths were measured in five randomly selected fields at 400x and then analyzed by Image J (National Institutes of Health, version 1.8.0).

### 2.13. Statistical Analysis

All statistical analyses were performed using IBM SPSS software (version 22.0; IBM Corp., Armonk, NY, USA). All data are presented as the mean ± standard deviation. Student’s t-test was used to compare the differences between two groups, and analysis of variance was applied for comparison among multiple groups. Linear regression was performed between the observed therapeutic targets. A *p*-value < 0.05 was considered statistically significant.

## 3. Results

### 3.1. Expression Levels of miR-140-3p and miR-574-3p Are Significantly Reduced in Placenta-Derived Exosomes from GDM

Our results revealed no difference in the concentration or number of placenta-derived exosomes collected from placental tissue between the normal (pregnancy) and GDM groups (Appendix A). Next-generation sequencing detected 38 medium- and high-abundance miRNAs with different expression levels in placenta-derived exosomes from GDM and normal pregnant women, including increased and decreased expression of 2 and of 36 miRNAs, respectively (Figure 1A and Appendix A). We validated the data of seven miRNAs with medium and high abundance that showed the greatest changes using qRT-PCR. The expression levels of placenta-derived exosomal miR-140-3p and miR-574-3p, which showed the greatest decreases, were evaluated (Figure 1C).

### 3.2. Placenta-Derived Exosomes from GDM Altered Endothelial Cell Function Depending on miR-140-3p and miR-574-3p In Vitro

Placenta-derived exosomes from GDM and normal women were transfected with miRNA agomir and antagomir, respectively, to investigate whether exosomes affect endothelial cell function via miR-140-3p and miR-574-3p, including proliferation, migration, and tube formation. Figure 2A shows the changes in expression of miR-140-3p and miR-574-3p in response to successful transfection with their agomir and antagomir, respectively. No differences were detected in the uptake of placenta-derived exosomes by HUVECs among the groups (Figure 2B). In contrast, the proliferation (Figure 2C), migration (Figure 2D), and total tube length (Figure 2E) of endothelial cells were reduced after treatment of placenta-derived exosomes from GDM women transfected with miR-140-3p or miR-574-3p agomir, compared with the negative control, whereas the normal group showed an increase in these parameters after treatment of the placenta-derived exosomes transfected with their antagomir.

### 3.3. Chemerin Reduces Placenta-Derived Exosomal miR-140-3p and miR-574-3p Expression by Inducing Placental Inflammation

Chemerin is an adipokine and an important regulator of angiogenesis, inflammation, blood pressure, and the immune system. Studies have shown that chemerin from peripheral blood, adipose tissue, and placenta tissue is elevated in GDM [19,31]. In this study, we noticed that the expression levels of miR-140-3p and miR-574-3p in placenta-derived exosomes were negatively correlated with chemerin in placental tissues in pregnant women (Figure 3A). Additionally, we established a chemerin-induced diabetic mouse model and knocked down chemR23, which is the receptor of chemerin, via a chemR23 shRNA lentivirus injection. Figure 3B shows that chemR23 in placental tissue changes in response to successful knockdown of chemR23. We observed that the levels of miR-140-3p and miR-574-3p were decreased in placenta-derived exosomes from chemerin-induced GDM mice, and increased in GDM mice with chemR23 knocked down (Figure 3C). This suggests that the reduced expression levels of exosomal miR-140-3p and miR-574-3p are associated with elevated chemerin in GDM.

Through immunofluorescence, we observed that macrophage infiltration was markedly exacerbated in GDM; co-localization of macrophages and chemerin was seen as well (Figure 4A). Previous research has shown that the aggregation of chemerin in fetal brain tissue leads to cognitive impairment in GDM offspring; chemerin buildup induces macrophage recruitment and pyroptosis, which is characterized by the activation of caspase-1 and secretion of pro-inflammatory cytokines such as IL-1β and IL-18 [20,32]. In this study, elevated levels of IL-18 and IL-1β were observed in the placental tissue of pregnant women with GDM (Figure 4B), which was observed in chemerin-induced diabetic mice as well; the expression levels of both decreased after knocking down chemR23 (Figure 4C). Moreover, the changes observed in the NOD-like receptor family pyrin domain containing 3 (NRLP3), apoptosis-associated speck-like protein containing CARD (Asc), and cleaved caspase 1 expression in placental macrophages were consistent with IL-18 and IL-1β (Figure 4D and Appendix A). These data confirm that chemerin induces macrophage recruitment as well as the release of inflammatory factors in placental tissue.

In order to further evaluate the role of macrophages and pro-inflammatory cytokines in chemerin-mediated miRNA changes, we isolated the macrophage supernatant from mouse placental tissue, co-cultured it with trophoblast cells, and then analyzed the trophoblast cell-derived exosomal miR-140-3p and miR-574-3p levels. Chemerin robustly increased the macrophage-induced decline of trophoblast cell-derived exosomal miR-140-3p and miR-574-3p expression, which was reversed by chemR23 knockdown (Figure 4E). Exosomal miR-140-3p and miR-574-3p expression was significantly inhibited in trophoblast cells treated by recombinant protein IL-18 and/or IL-1β compared with the control group (Figure 4F). Additionally, after successful silencing of IL-18 and/or IL-1β in macrophages (Figure 4G), the chemerin-induced macrophage-mediated reduction of exosomal miR-140-3p and miR-574-3p expression was reversed (Figure 4H). IL-18 and/or IL-1β decreased exosomal miR-140-3p and miR-574-3p expression by inhibiting primary miRNA processing in trophoblast cells (Appendix A). Collectively, these results demonstrate that chemerin reduces placenta-derived exosomal miR-140-3p and miR-574-3p expression through the recruitment of macrophages and the release of IL-18 and IL-1β.

### 3.4. MiR-140-3p and miR-574-3p Down-Regulate VEGF Expression in Endothelial Cells

Vascular endothelial growth factor (VEGF) is a diffusible endothelial cell-specific mitogen that mediates endothelial cell survival, migration, and proliferation during angiogenesis. Our results show that VEGF expression was significantly higher in the umbilical vein tissue of GDM patients (Figure 5A) and was negatively associated with miR-140-3p and miR-574-3p levels in umbilical vein tissue (Figure 5B). Therefore, we hypothesize that miR-140-3p and miR-574-3p affect endothelial cell function by regulating VEGF expression. miR-140-3p and miR-574-3p agomir reduced VEGF expression in vitro, while their antagomir had the opposite effect (Figure 5C).

Through miRNAs target gene prediction software and protein interaction analysis, we found that miR-140-3p has multiple target genes and revealed more interactions between VEGF and other potential target genes (Figure 5D). 3′-untranslated regions (3′UTRs) are the noncoding parts of mRNAs that regulate mRNA degradation, translation, and localization. We constructed wild-type and mutant luciferase reporter plasmids in the binding site in the 3’UTR of VEGF. miR-140-3p reduced the luciferase activity of wild-type VEGF 3’UTR, and had no effect on its mutant (Figure 5E).

miR-574-3p binds to heterogeneous nuclear ribonucleoprotein L (hnRNP L) and prevents hnRNP L from binding to the VEGFA mRNA, ultimately reducing VEGFA mRNA translation [33]. To clarify whether the mediating effect of miR-574-3p on the expression of VEGF depends on hnRNP L, an hnRNP L OE plasmid was constructed and hnRNP L and VEGF levels were determined by Western blotting. We observed an increase in hnRNP L OE plasmid levels and blockade of the inhibitory effect of miR-574-3p on VEGF expression (Figure 5F). Moreover, OE of hnRNP L reversed the impairments in proliferation, migration, and tube formation caused by the miR-574-3p agomir (Appendix A). Taken together, our results showed that miR-574-3p downregulates VEGF expression by competitively binding hnRNP L, thereby inhibiting the functional capacity of umbilical vein endothelial cells.

## 4. Discussion

In this report, we have demonstrated that chemerin reduces placenta-derived exosomal miR-140-3p and miR-574-3p levels by inducing placental inflammation, thereby promoting the proliferation, migration, and tube formation of umbilical vein endothelial cells in GDM by upregulating VEGF expression.

GDM is a common metabolic disorder the prevalence of which is increasing in line with the increasing incidence of obesity and trend towards older maternal age seen worldwide. As an important mediator of the exchange of oxygen and nutrients between mother and fetus, the placenta plays an important role in fetal growth as well as the development of complications in offspring. Studies have found that placental abnormalities in GDM include an increased immature villous, increased placental weight, and increased measures of angiogenesis [34]. Fetoplacental endothelial dysfunction in GDM has been observed for decades, and plays a crucial role in offspring complications. However, the mechanism has not been elucidated to date. Studies have shown that fetoplacental endothelial dysfunction in GDM is related to the increased release of exosomes induced by hyperglycemia, hyperinsulinemia, and oxidative stress [23,35]. The concentration and bioactivity of placenta-derived exosomes present in maternal plasma were higher in GDM patients compared with those having a normal pregnancy, modulating the release of proinflammatory cytokines from endothelial cells [22]. Herein, we isolated placenta-derived exosomes from placenta tissue and found no difference in the concentration of placenta-derived exosomes between normal and GDM pregnant women. Placenta-derived exosomes from GDM women promoted the proliferation, migration, and tube formation of endothelial cells compared with those from normal women. Thus, fetoplacental endothelial dysfunction in GDM appears to be associated with changes in the bioactivity of placenta-derived exosomes. In addition, GDM along with failure of endothelial cells vasodilation since insulin resistance might result in blunted endothelium-derived NO-dependent vascular response to this hormone; this needs to be investigated further [36,37,38].

Placenta-derived exosomal miRNAs are released into the maternal circulation during pregnancy, and levels of a variety of miRNAs change in GDM patients [39]. Herein, we have shown an association of placenta-derived exosomal miRNA changes with the umbilical vein endothelial dysfunction seen in GDM. In addition, we have shown that placenta-derived exosomal miR-140-3p and miR-574-3p are significantly reduced in GDM patients. Previous studies on miR-140-3p and miR-574-3p have focused mainly on suppressing the proliferation, migration, and invasion of tumor cells [40,41,42,43]. Increased expression of miR-140-3p has been found to induce hypoxia reoxygenation-stimulated vascular endothelial cell injury by inhibiting proliferation and function [44]. Whether the fetoplacental endothelial dysfunction seen in GDM is relevant to the reduced expression of miR-140-3p and miR-574-3p has not been elucidated. We first demonstrated that placenta-derived exosomal miR-140-3p and miR-574-3p suppress the proliferation, migration, and tube information of umbilical vein endothelial cells. Despite reports that miR-140 decreases VEGF expression [44,45,46], the underlying mechanism remains unclear. Similarly, we observed that miR-140-3p decreased VEGF expression in endothelial cells. miR-140-3p regulates the expression of ZO-1-associated kinase (ZAK) and Forkhead box protein Q1 (FOXQ1) by targeting their mRNA 3′UTR [41,47]. In this investigation, we found that miR-140-3p binds to the 3′UTR of VEGF mRNA, which suggests that miR-140-3p reduces VEGF expression by binding directly to VEGF mRNA. MiR-574-3p regulates VEGFA translation by competitively binding hnRNP L [33]. Moreover, our results show that miR-574-3p downregulates VEGF expression by targeting hnRNP L. miR-140-3p and miR-574-3p combined treatment inhibited VEGF expression in umbilical vein endothelial cells more significantly than miR-140-3p or miR-574-3p alone; thus, the reduced levels of miR-140-3p and miR-574-3p offset the inhibition of VEGF expression, resulting in placental-derived exosome-mediated fetoplacental endothelial dysfunction in GDM.

Chemerin, a newly discovered adipocytokine, is involved in metabolic diseases and inflammation regulation. It promotes the chemotaxis of macrophages and immature dendritic cells in a chemR23-dependent manner [48]. Studies have shown that chemerin expression is increased in GDM [27,49]. We previously found that chemerin reduced the number of neurons by inducing macrophage recruitment, activation of pyroptosis, and the release of inflammatory cytokines in the fetal brain tissue of offspring of diabetic dams, in turn leading to cognitive impairment [20]. However, it is not clear whether chemerin is associated with placental inflammation in GDM. Additionally, the NRLP3 inflammasome may participate in this process, the assembly of which results in the caspase 1-dependent release of IL-1β and IL-18 [50]. This study confirmed that chemerin recruited macrophages and promoted NLRP3 expression, caspase 1 activation, and the release of inflammatory factors IL-1β and IL-18 in the placental tissue of GDM, thereby inducing placental inflammation. Few studies have shown the interaction between chemerin and miRNAs, and one has reported that chemerin decreased miR-217 expression and promoted proliferation, adipogenesis, and angiogenesis in 3T3-L1 preadipocytes [51]. Several groups have reported that exosomal miRNAs are regulated by proinflammatory cytokines [52,53]. We found that trophoblast cell-derived exosomal miR-140-3p and miR-574-3p were strongly downregulated by the supernatant from macrophages treated with chemerin plus IL-18 and/or IL-1β stimulation. Meanwhile, the levels of placenta-derived exosomal miR-140-3p and miR-574-3p were negatively related to chemerin from clinical samples. This suggests that chemerin inhibits trophoblast cell-derived exosomal miR-140-3p and miR-574-3p expression by inducing placental inflammation. 

This study has several limitations, including the relatively low number of subjects, inclusion of subjects with a relatively high maternal age, and smaller gestational weight gain of the GDM group relative to the control. Second, chemerin-induced placental macrophage recruitment was explored only in the placenta tissue of pregnant women by immunofluorescence; ideally, these results should be verified in animal experiments. Thus, additional studies are needed to reveal the specific mechanism underlying chemerin-induced placental inflammation. Third, we found that IL-1β and IL-18 reduced the expression of exosomal miR-140-3p, miR-574-3p, and pri-miRNA processing-related proteins (including Drosha, DGCR8, and Exportin-5) in trophoblast cells, which suggests that IL-1β and IL-18 alter trophoblast cell-derived exosomal miR-140-3p and miR-574-3p expression by regulating trophoblast pri-miR-140-3p and pri-miR-574-3p processing and nuclear export. However, more experiments are needed to confirm this. In addition, IL-1β and IL-18 had no effect on the expression levels of pri-miR-140-3p and pri-miR-574-3p; whether they are related to the reduced transcriptional activity of miR-140-3p and miR-574-3p or accumulation of nuclear pri-miR-140-3p and pri-miR-574-3p induced by processing blockade requires further study. 

In conclusion, chemerin reduced placenta-derived exosomal miR-140-3p and miR-574-3p levels by inducing placental recruitment of macrophages and the release of inflammatory factors, thereby increasing VEGF expression and promoting the proliferation, migration, and tube formation of umbilical vein endothelial cells in the GDM group. Our results indicate that placenta-derived exosomal miR-140-3p and miR-574-3p have potential as markers of GDM and that their agomirs could reduce the incidence of offspring complications by attenuating the function of umbilical vein endothelial cells.

## Figures and Tables

**Figure 1 cells-11-03457-f001:**
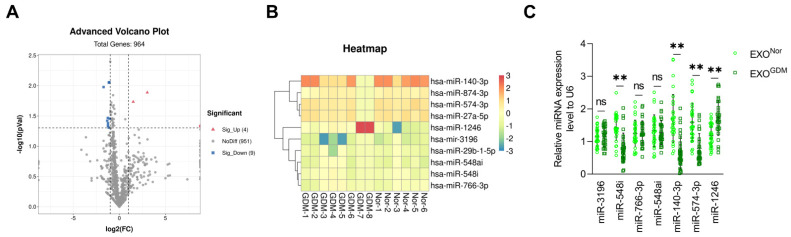
GDM differentially-expressed miRNAs. (**A**) Volcano plots show the fold change (log2) against the *p* value (−log10) of normal (n = 6) and GDM (n = 8) pregnant women. The significantly upregulated miRNAs (FC > 2) are depicted in red and the downregulated miRNAs (FC < 0.5) in blue. (**B**) Heatmap of the ten most changed miRNAs with medium and high abundance. (**C**) Scatter plot showing relative expression of the seven most changed miRNAs with medium and high abundance in placenta-derived exosomes from GDM (n = 30) versus those from normal (n = 30) pregnant women by qRT-PCR. ** *p* < 0.01, ns: no significance.

**Figure 2 cells-11-03457-f002:**
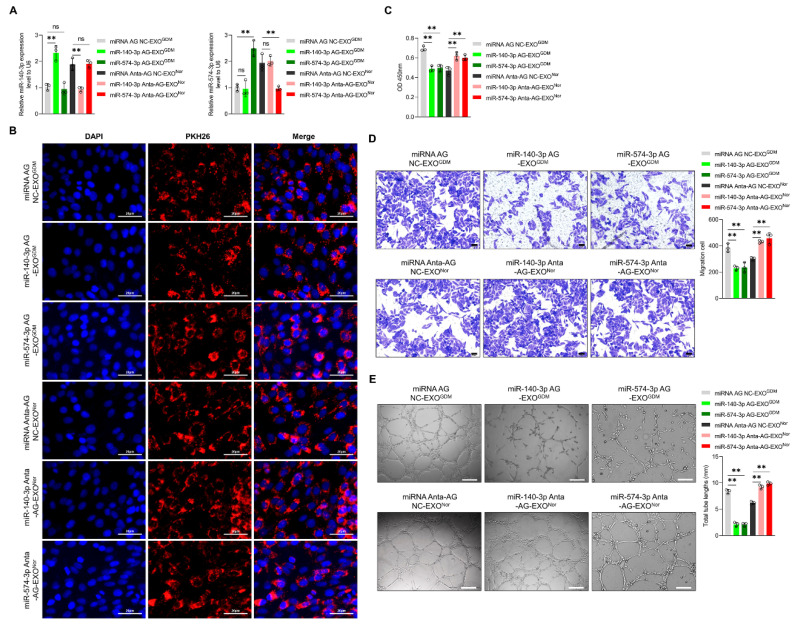
Effects of transfection of placenta-derived exosomes from normal and GDM women with miR-140-3p or miR-574-3p agomir or antagomir on the proliferation, migration, and tube formation of HUVECs in vitro. (**A**) Verification of placenta-derived exosomal miR-140-3p and miR-574-3p expression by qRT-PCR. EXO^GDM^ transfected with miRNA agomir negative control (miRNA AG NC), miR-140-3p agomir (miR-140-3p AG), or miR-574-3p agomir (miR-574-3p AG). EXO^Nor^ were transfected with miRNA antagomir negative control (miRNA Anta-AG NC), miR-140-3p antagomir (miR-140-3p Anta-AG), or miR-574-3p antagomir (miR-574-3p Anta-AG). (**B**) Immunofluorescence staining for placenta-derived exosomes marked with PKH26, (Scale bar: 20 µm). (**C**) The CCK-8 assay was applied to detect the proliferation of HUVECs after treatment with miRNA AG NC-EXO^GDM^, miR-140-3p AG-EXO^GDM^, miR-574-3p AG-EXO^GDM^, miRNA Anta-AG NC-EXO^Nor^, miR-140-3p Anta-AG-EXO^Nor^, or miR-574-3p Anta-AG-EXO^Nor^. (**D**) A transwell assay was applied to detect the migration of HUVECs after different treatments, (Scale bar: 20 µm). (**E**) A tube formation assay was applied to detect the tube formation of HUVECs after different treatments (n = 3, Scale bar: 100 µm). ** *p* < 0.01. ns: no significance.

**Figure 3 cells-11-03457-f003:**
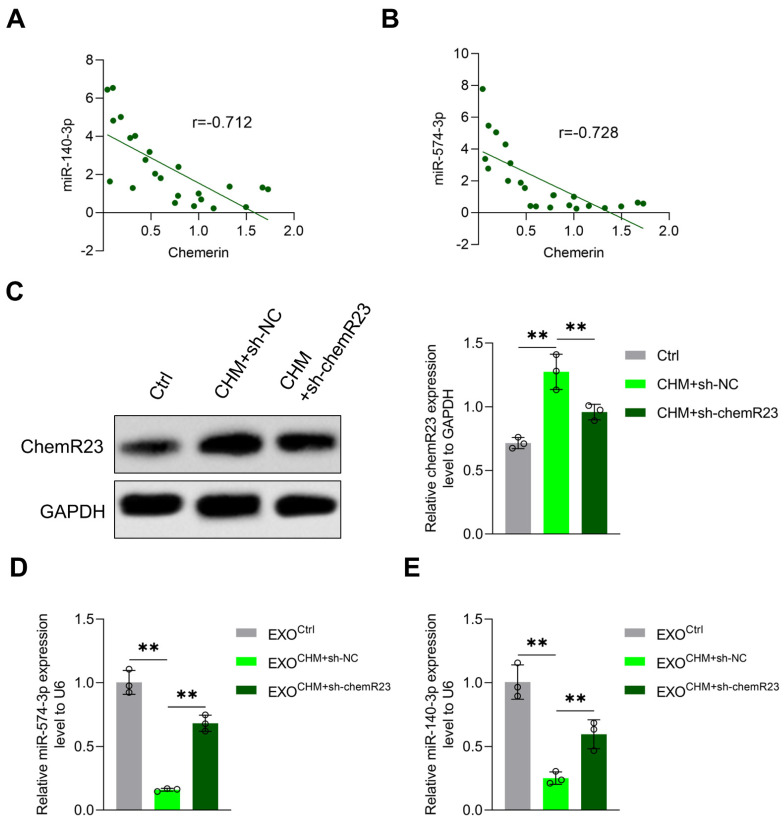
Placenta-derived exosomal miR-140-3p and miR-574-3p expression was negatively regulated by chemerin. (**A**) Linear regression analysis of placenta-derived exosomal miR-140-3p expression and chemerin content in placenta tissues in pregnant women (n = 23). (**B**) Linear regression analysis of placenta-derived exosomal miR-574-3p expression and chemerin content in placenta tissues in pregnant women (n = 23). (**C**) ChemR23 protein levels in placenta tissues from control mice (Ctrl), chemerin-induced diabetic mice with injection of shRNA-negative control (CHM+sh-NC), and chemerin-induced diabetic mice with injection of chemR23 shRNA (CHM+sh-chemR23) detected by western blotting (n = 3). (**D**) Relative miR-140-3p and (**E**) miR-574-3p expression of placenta-derived exosomes from Control (EXO^Ctrl^), CHM+sh-NC (EXO^CHM+sh-NC^) and CHM+sh-chemR23 (EXO^CHM+sh-chemR23^) mice measured with qRT-PCR (n = 3). ** *p* < 0.01.

**Figure 4 cells-11-03457-f004:**
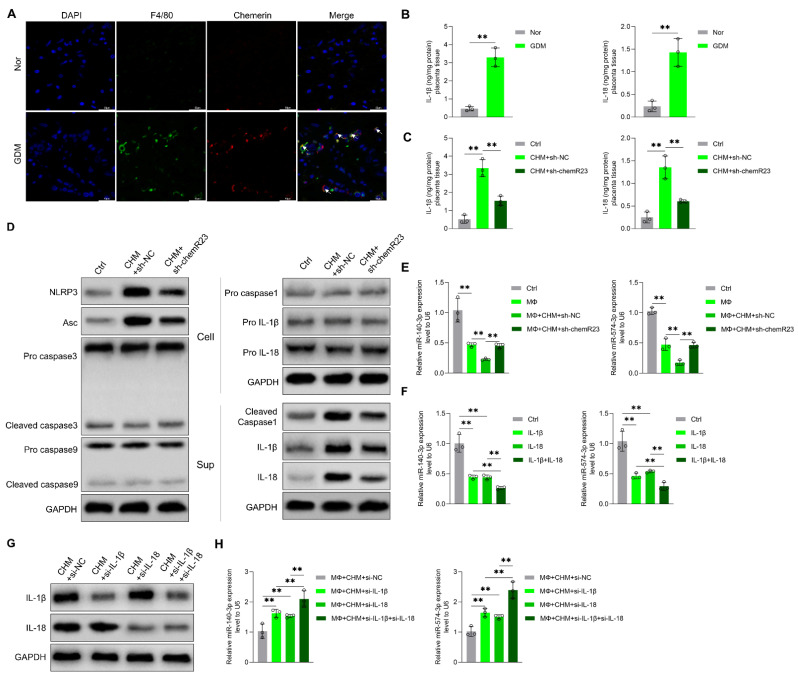
Chemerin-induced placental inflammation-regulated trophoblast cell-derived exosomal miR-140-3p and miR-574-3p expression. (**A**) Immunofluorescence staining for F4/80 (macrophages) and chemerin of placenta tissue in normal and GDM pregnant women. (**B**) Content of IL-18 and IL-1β in placenta tissue of normal and GDM pregnant women was measured by ELISA (n = 3). (**C**) Content of IL-18 and IL-1β in placenta tissue of Ctrl, CHM+sh-NC, and CHM+sh-chemR23 mice was measured by ELISA (n = 3). (**D**) Protein levels of NOD-like receptor family pyrin domain containing 3 (NRLP3), apoptosis-associated speck-like protein containing CARD (Asc), pro caspase 3, cleaved caspase 3, pro caspase 9, cleaved caspase 9, pro caspase 1, cleaved caspase 1, pro-IL-1β, and pro-IL-18 in placental macrophages were measured by Western blotting. Protein levels of cleaved caspase 1, IL-1β, and IL-18 in the culture supernatants of placental macrophages were measured by Western blotting as well. Glyceraldehyde-3-phosphate dehydrogenase (GAPDH) served as the internal control (n = 3). (**E**) Trophoblast cells were co-incubated with control (Ctrl), the supernatant from macrophage, macrophage+CHM+sh-NC, and macrophage+CHM+sh-chemR23 for 24 h, then relative exosomal miR-140-3p and miR-574-3p expression was measured by qRT-PCR (n = 3). (**F**) Trophoblast cells were treated with Control, recombinant protein IL-18, recombinant protein IL-1β, and recombinant protein IL-18+IL-1β. Relative exosomal miR-140-3p and miR-574-3p expression was measured by qRT-PCR (n = 3). (**G**) Changed protein levels of IL-18 and IL-1β in macrophage after treatment with chemerin and siRNA NC (CHM+si-NC), IL-1β siRNA (CHM+si-IL-1β), IL-18 siRNA (CHM+si-IL-18) and IL-1β siRNA and IL-18 siRNA (CHM+si-IL-1β+si-IL-18) were measured by Western blotting (n = 3). (**H**) Trophoblast cells were co-incubated with the supernatant of macrophage+CHM+si-NC, macrophage+CHM+si-IL-1β, macrophage+CHM+si-IL-18, and macrophage+CHM+si-IL-1β+si-IL-18, then relative exosomal miR-140-3p and miR-574-3p expression was measured by qRT-PCR (n = 3). ** *p* < 0.01. Scale bar: 50 μm.

**Figure 5 cells-11-03457-f005:**
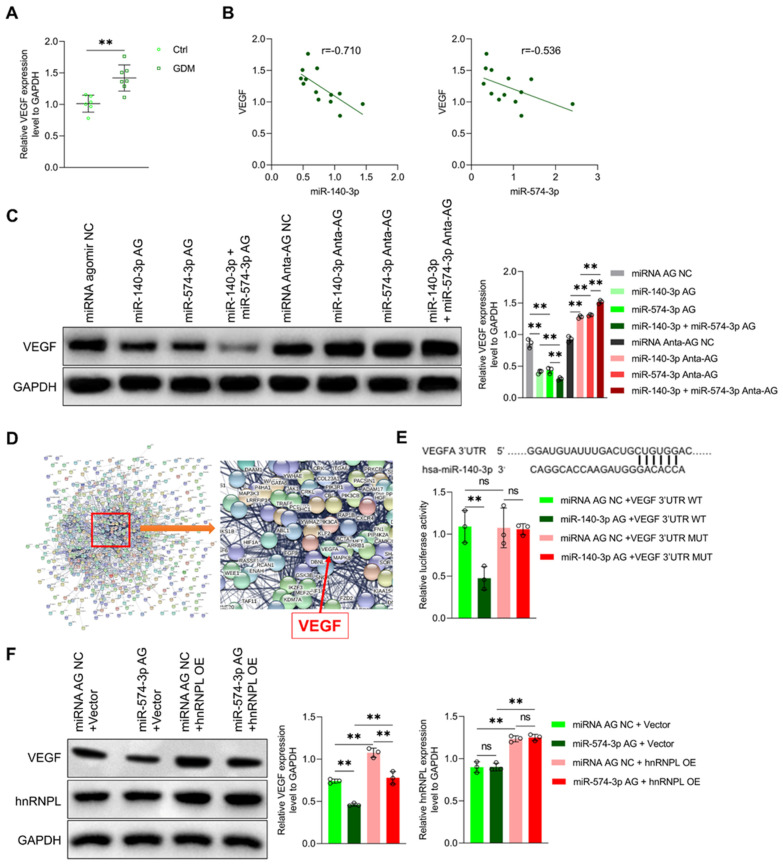
miR-140-3p and miR-574-3p downregulate VEGF expression. (**A**) VEGF expression in the umbilical vein tissue of normal (n = 6) and GDM (n = 7) pregnant women was measured by Western blotting. (**B**) Linear regression analysis of miR-140-3p or miR-574-3p expression and VEGF in the umbilical vein tissue in pregnant women (n = 13). (**C**) VEGF expression of MUVECs after treatment with miRNA AG NC, miR-140-3p AG, miR-574-3p AG, miR-140-3p + miR-574-3p AG, miRNA Anta-AG NC, miR-140-3p Anta-AG, miR-574-3p Anta-AG, and miR-140-3p + miR-574-3p Anta-AG was measured by Western blotting (n = 3). (**D**) TargetScan and a protein–protein interaction (PPI) network were applied to identify the targets of miR-140-3p and their interaction. (**E**) Co-transfection of wild-type or mutant seed regions of VEGF 3′UTR were constructed with miRNA AG NC or miR-140-3p AG in MUVECs. A luciferase assay was applied to detect the luciferase activity (n = 3). (**F**) Protein levels of VEGF and hnRNP L after transfection with miRNA AG NC+vector and miR-574-3p AG+vector, miRNA AG NC+hnRNP L overexpression (OE), and miR-574-3p AG+hnRNPL OE were measured by Western blotting (n = 3). ** *p* < 0.01. ns: no significance.

## Data Availability

The next gene sequencing results can be found at GEO (https://www.ncbi.nlm.nih.gov/geo/query/acc.cgi?acc=GSE213799) (accessed on 27 July 2022).

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
