# Peer review of "Chemerin-Induced Down-Regulation of Placenta-Derived Exosomal miR-140-3p and miR-574-3p Promotes Umbilical Vein Endothelial Cells Proliferation, Migration, and Tube Formation in Gestational Diabetes Mellitus"

_cells, 2022, doi:10.3390/cells11213457_

Round 1
Reviewer 1 Report
The study provides clear evidence of a connection between exosomal miRNA secretion and the impact on endothelium in the context of GDM pregnancy. As it stands the story seems complete. My concern is this story seems complete and self contained. But there are other aspects to consider in the physiology that are not described because the authors are so fixed on just two cytokines and the assumption the negative impact of GDM is through these miRNAs. I would like to see in the discussion some more recognition that endothelial function is always maintained through a balance of angiogenesis and functional vasodilation. Also that in pregnancy failure to vasodilate may be through an early failure of angiogenesis or a late failure of vasodilation. This model suggest the role of miRNA is in mediating failures of angiogenesis but it is not clear how much that actually contributes to the symptoms of GDM. All the data presented is from experiments designed to infer that miRNA is key, but does it actually happen to a great extend in human pregnancy? Or is it just one of many processes and may be more important in some patients than others. Example- there is plenty of evidecne that other cytokines mediate damage to HUVEC etc through ROS, and that has nothing to do with this axis of control. And are IL18 and IL1B the only cytokines expected to mediate this effect? How universal is the observation that these cytokines are disturbed in GDM subjects? What other factors are disturbed that could mediate these effects in humans? What other factors are disturbed that could mediate effects independently of miRNA. How do these two pathways to disfunction relate to each other in the real world?
Author Response
Thanks for your comments. we have uploaded the response as a word file. Please see the attachment.

Reviewer 2 Report
This paper presents evidence for a relationship between Gestational Diabetes, reduced placental release of miR-140-3p and miR-574-3p in extracellular vesicles, and enhanced activation of endothelial cells via VEGF. This is a worthwhile addition to the literature.
My major concerns, as detailed below, are a serious lack of experimental detail (5), insufficient evidence that exosomes are being measured (3), and that the confounding effects of chemerin on blood glucose and insulin production make it impossible to draw any conclusions from the mouse model about whether chemerin itself regulates any of the endpoints studied (miRNA, macrophages etc) or it’s the diabetes causing these outcomes (12). The data for a chemerin association with these miRNAs is weak.
1) The paragraph in the introduction (starting line 57) could do a better job situating the current study within the literature on extracellular vesicles and their contents in gestational diabetes. The present study still contributes new information to the field, but there are many published studies of extracellular vesicles in GDM. Similarly, it would be helpful, especially for readers not familiar with it, to introduce a few more details of what is already known about chemerin expression in gestational diabetes and its relationship to abnormal placental angiogenesis (mostly from pre-eclampsia ).
2) There’s no mention of the macrophage/cytokine experiments in the introduction or a rationale given for performing these. It’s not clear how they fit into the question being addressed. Is placental inflammation a feature of GDM?
3) The exosome isolation method used here will greatly enrich for exosomes, but not completely exclude microvesicles. It would be more accurate to refer to it as extracellular vesicles, or an exosome-enriched extracellular vesicle preparation. Figure S2 D shows a large peak that’s definitely within exosome range, but some larger extracellular vesicles as well. Need references for exosome markers used, and a control fraction (medium before isolation or before final spin or filtration step) to be able to assess degree of enrichment. If there was serum used in the culture medium, the medium should also be analyzed for comparison, as serum contains extracellular vesicles (PMC7830136). Ideally, also quantifying a microvesicle marker for comparison would help. Finally, the EM does not appear to show the cup shape characteristic of exosomes, again suggesting other types of extracellular vesicles.
4) Materials and Methods section needs much more detail so that experiments can be replicated by others, and evaluated by readers and reviewers.
· Suppliers/source should be specified for all reagents and kits
· Concentrations and times should be given for all experiments (e.g. cytokine treatments of cells, western blotting blocking & secondary ab steps; how much protein loaded in ELISAs and westerns; how were protein concentrations determined).
· How were sample sizes determined? Sample size/replicate numbers should be listed for all experiments.
· Specify how and when in pregnancy glucose testing was performed (two step or one step test, how much glucose), as well as how GDM and abnormal glucose levels were defined (reference to IADPSG standards would work for latter). What was mode of delivery for placentas?
· The streptozotocin model described as “typical diabetic model” should be referenced.
· What was the rationale for the dose of chemerin, and when was it administered? How was fasting blood glucose assessed (how long was fast, how was blood taken, how was glucose measured), how was insulin measured, and how was HOMA-IR calculated? How many animals were tested?
· Specify culture medium, well size and size of placental explants
· How were macrophages obtained- was it the same placentas as the exosomes? and how long and at what density were they cultured for supernatant collection?
· Where did trophoblast cells for cultures come from/ what are they?
· How was reverse transcription performed? How were primers validated – were melting curve analysis and serial dilutions performed? Was the housekeeping gene tested to make sure it does not change with the various treatments? What were the cycling conditions? (note- the RT in qPCR does not stand for Real-Time, it stands for reverse transcription).
· Where was the overexpression plasmid and vector obtained and how much was used for how long, with what transfection method?
· Need description of electron microscopy methods, and how many samples were analyzed.
· There is no description at all of next gen sequencing or analysis methods.
· Description of exosome transfection procedure (loading agromir/antagomir) is needed
· How many images were analyzed for transwell and tube formation assays and how was counting/measuring performed?
5) The next gen sequencing results should be made available in a public repository and the link provided. A little more summary of overall results, for example how many miRNA were detected total, would be helpful.
6) The number of data points in figure 1C does not match the description of the sample size.
7) In figures 2C,D and E, were comparisons made between GDM and Normal exosomes without agomir/antagomir? lines 253-254 suggest that these were compared but I don’t see it in the graphs. This sentence also suggests that untransfected exosomes from GDM and controls were compared to HUVEC untreated with exosomes, but no such data are shown.
8) Characterization of the chemerin-induced diabetes model should include glucose tolerance testing. Fasting glucose and glucose tolerance should also be measured in the chemerin-induced, chemR23 treated mice.
9) The evidence for a link between chemerin and the miRNAs is weak. The correlation in Figure 3A is highly suspect. It seems to be driven entirely by 2 placentas with low chemerin, with no other dose dependency visible. 3B is not much stronger. At what day were westerns performed in the mice? And how were the mouse placenta exosomes collected – no description is given
10) It's not possible to make out cell locations/ morphology in figure 4A. A light microscope image might help. The ELISA results are given in pg/mL – because it’s tissue it should be normalized to mg of tissue or protein, not mL.
11) Line 472-475 implies that the macrophages/inflammation is responsible for the change in VEGF, but this is not shown by the results, which instead show direct regulation of VEGF by the miRNA. Discussion cites one previous study of miR-140 link to VEGF and angiogenesis, but there are several more.
12) The paper concludes that, because treating mice with chemerin reduces miR-140-3p and miR574-3p, changes placental macrophages etc, that chemerin regulates these processes. This cannot be concluded because the chemerin treatment induces diabetes, which might be responsible for the changes to the placenta.
13) The discussion could explain in more detail how a reduction at term in GDM of miRNAs that inhibit angiogenesis explains the complex endothelial dysfunction observed in GDM. GDM placentas tend to show less branched villi, with more centrally located capillaries, fewer vasculosyncytial membranes, and greater capillary density, along with more fibrinoid deposition. It would be helpful to explain how this relates to the central findings.
Typos etc.
Typo in “Supplementary Materials” heading
In figure 1 legend, I think it should say “ten most changed” not “ten mostly changed”
Supplemental table 1. Use more digits or scientific notation to show p-values for 1h and 2h glucose loads
Figure 3 legend does not describe panels D and E
Graph text is almost too small to read in most panels. Figure 5D is illegible
Referring to treatment with an siRNA for the chemerin receptor as “chemerin treatment” is confusing, since there is also a group treated with chemerin.
Author Response
Thanks for your comments. we have uploaded the responses as a word file. Please see the attachment.
